# Anti-Tumor Effects of Cecropin A and Drosocin Incorporated into Macrophage-like Cells Against Hematopoietic Tumors in *Drosophila mxc* Mutants

**DOI:** 10.3390/cells14060389

**Published:** 2025-03-07

**Authors:** Marina Hirata, Tadashi Nomura, Yoshihiro H. Inoue

**Affiliations:** 1Biomedical Research Center, Kyoto Institute of Technology, Kyoto 606-0962, Japantadnom@kit.ac.jp (T.N.); 2Graduate School of Science and Technology, Kyoto Institute of Technology, Kyoto 606-8585, Japan

**Keywords:** *Drosophila* tumor, hemocytes, apoptosis, antimicrobial peptides (AMPs), phosphatidylserine, endocytosis

## Abstract

Five major antimicrobial peptides (AMPs) in *Drosophila* are induced in *multiple sex combs* (*mxc*) mutant larvae harboring lymph gland (LG) tumors, and they exhibit anti-tumor effects. The effects of other well-known AMPs, Cecropin A and Drosocin, remain unexplored. We investigated the tumor-elimination mechanism of these AMPs. A half-dose reduction in either the *Toll* or *Imd* gene reduced the induction of these AMPs and enhanced tumor growth in *mxc^mbn1^* mutant larvae, indicating that their anti-tumor effects depend on the innate immune pathway. Overexpression of these AMPs in the fat body suppressed tumor growth without affecting cell proliferation. Apoptosis was promoted in the mutant but not in normal LGs. Conversely, knockdown of them inhibited apoptosis and enhanced tumor growth; therefore, they inhibit LG tumor growth by inducing apoptosis. The AMPs from the fat body were incorporated into the hemocytes of mutant but not normal larvae. Another AMP, Drosomycin, was taken up via phagocytosis factors. Enhanced phosphatidylserine signals were observed on the tumor surface. Inhibition of the signals exposed on the cell surface enhanced tumor growth. AMPs may target phosphatidylserine in tumors to induce apoptosis and execute their tumor-specific effects. AMPs could be beneficial anti-cancer drugs with minimal side effects for clinical development.

## 1. Introduction

Insects such as *Drosophila* do not possess acquired immunity and thus rely on innate immunity for protection [1]. Although innate immunity provides initial defense only, its involvement in initiating and regulating acquired immunity was recently reaffirmed [2,3]. Many studies on the molecular mechanisms of innate immunity have been conducted using *Drosophila* because its specific functions are not overshadowed by the more powerful acquired immunity [4]. Thanks to advanced genetic and developmental biology methods, *Drosophila* offers an excellent model for studying immunity.

In *Drosophila*, innate immunity includes humoral and cellular defense responses [5,6,7]. Antimicrobial peptides (AMPs) play a major role in the humoral defense response. In response to infection, AMPs are produced by the fat body (FB), the functions of which are similar to those of the mammalian liver and adipose tissue [8]. AMP production is mediated by the activation of either, or both, of two major signaling pathways: Toll and Imd [1,8,9]. The Toll pathway is activated primarily by Gram-positive bacteria or fungal infections [10,11]. Recognition proteins identify cell wall components common to those bacteria or fungi [12,13]. A serine protease cascade is then activated, ultimately producing an active Spätzle [14]. This binds to the Toll receptor, and the signals are transmitted into the cytoplasm [15]. Subsequently, the degradation of Cactus—which inhibits the nuclear translocation of the Dif and Dorsal transcription factors—translocates the factors into the nucleus and induces the transcription of relevant AMP genes [8,14]. In contrast, the Imd pathway is activated mainly by Gram-negative bacterial infection [10]. The transmembrane receptor recognizes cell wall components common to these bacteria [16,17]. It then transmits signals via protein complexes, including Imd [18,19]. Eventually, the Relish transcription factor allows for its nuclear translocation and the induction of the relevant AMP genes’ transcription [16,17]. The Toll- and Imd-mediated pathways in *Drosophila* are highly homologous to the mammalian Toll-like receptor and tumor necrosis factor (TNF) receptor-mediated signaling pathways, respectively [6,18,20]. Some of their target gene products, AMPs, are also conserved among different species. The first AMP, cecropin, was isolated in the 1980s from the silkworm moth, *Hyalophora cecropia* [21]. The following seven major AMPs are well characterized in *Drosophila*: Attacin, Cecropin, Defensin, Diptericin, Drosocin, Drosomycin, and Metchinikowin [1]. Among them, synthetic cecropin A peptides have been shown to demonstrate anti-tumor effects against cancer cells in culture systems [22,23,24]. AMP’s cytotoxic and tumor growth-suppressing properties have been demonstrated in vitro as well as in *Drosophila* bodies [25,26,27,28]. Five of these seven AMPs are induced via the activation of the innate immune pathway in response to tumors arising in imaginal discs and hematopoietic tissues, and effectively suppress tumor growth by inducing apoptosis in *Drosophila* [25,28]. However, it is still unclear how innate immunity recognizes tumor cells, how the two innate immune pathways are activated to induce AMPs, and how tumors are suppressed by AMPs from the FB.

In *Drosophila*, mature hemocytes circulating in the hemolymph play a key role in important innate immunity responses [29]. In the latter larval stage, hemocytes are supplied from the hematopoietic pockets and lymph gland (LG). Plasmatocytes comprise approximately 95% of hemocytes and act like macrophages, which eliminate apoptotic cells via phagocytosis. Hemocytes play a role in fighting bacterial infections and tumor cells by conveying immune signals toward the FB to induce the expression of AMPs [30,31]. The transcriptional regulation of Drosocin via inter-tissue communication by hemocytes was characterized well recently [30]. In *Drosophila*, hemocyte development and function are very similar to those of mammalian macrophages. Immune cell recruitment to tumor-forming foci is a hallmark of cancer [32]. *Drosophila* models also showed that hemocytes accumulate in tumors when they recognize damage to the basement membrane [33,34,35]. These cells produce Spätzle in tumors arising in the imaginal discs [27]. However, the detailed mechanisms of tumor recognition, signaling, and subsequent tumor suppression by hemocytes remain unelucidated.

*mxc^mbn1^* is a loss-of-function allele of the *multi sex combs* (*mxc*) gene; hemizygotes for the mutation demonstrate enlarged LGs in the larval stage [25,36]. Moreover, *mxc^mbn1^* mutants exhibit a leukemia-like phenotype, with increased numbers of undifferentiated hematopoietic cells in the hemolymph, which invade other tissues [37,38]. LG tumors in *mxc^mbn1^* mutants exhibit overgrowth and invasive metastasis. In the mutant larvae, five of the seven major AMPs are induced and exhibit anti-tumor effects [25]. By contrast, two remaining AMPs, Cecropin A and Drosocin, have not yet been analyzed, although they are believed to be related to cancer and the innate immune system, as previously mentioned.

In this study, we focused on these two AMPs to determine whether they are induced in response to LG tumors and possess tumor-suppressive effects. We began by verifying whether these AMP genes are induced in the FB of *mxc^mbn1^* mutants. We then examined whether the AMPs exhibit inhibitory effects on LG tumors and suppress tumor growth by inducing apoptosis, as observed in the other five AMPs. Further, we investigated the mechanism of their anti-tumor effect in a tumor-specific manner. Our findings are expected to elucidate the mechanisms of innate immune system activation and tumor suppression in response to tumors in *Drosophila*. Although mammalian AMPs exhibit anti-cancer potential [39], most studies on their effects have been conducted in cultured cells. In contrast, our findings were determined using living organisms. This may benefit the future development of AMPs as promising new anti-cancer drugs with minimal side effects.

## 2. Materials and Methods

### 2.1. Drosophila Stocks

*w^1118^* (*w*) was used as a normal control stock. The recessive lethal allele of *mxc*, *mxc^mbn1^*, was used as a malignant hematopoietic tumor mutant [25,36,37,38]. The following Gal4 driver stocks were used for ectopic expression in specific larval tissues or cells as follows: *w**; *P{w[+mC]=r4-GAL4}3* for the induction of gene expression in the FB (#33832; Bloomington Drosophila Stock Center (BDRC) [25]), *P{He-GAL4.Z}85* (*He-Gal4*) (#8700; BDSC) for moderate induction in circulating hemocytes [36], and *P{upd3-GAL4}* (a gift from N. Perrimon, Harvard Medical School, Boston, MA, USA) for the induction of gene expression in the LG tumor of *mxc^mbn1^* mutants. *Dro-GFP* (a gift from M. Miura, University of Tokyo, Tokyo, Japan [39]), *CecA1-GFP* (#600216; BDSC [39]), and *Drs-YFP* (a gift from Y. Yagi, Nagoya University, Nagoya, Japan [40]) were used to monitor the gene expression of the *Dro*, *CecA* and *Drs* genes, respectively. To visualize the localization of Drosomycin in the circulating hemocytes, *P{Drs-GFP.JM804}1* stock (#55707; BDSC) was used, in which the EGFP-tagged Drosomycin is expressed under its gene promoter. For dsRNA-dependent gene silencing, the following *UAS-RNAi* stocks were used: *P{GD498}v42503* (*UAS-DroRNAi*) (#42503; VDRC) [30], *P{GD3965}v9710* (*UAS-CecA1RNAi*) (#9710; VDRC) [41], *P{GD15912}* (*UAS-xkrRNAi*) (#48383; VDRC) [42], *P{w[+mC]=UAS-drpr.dsRNA}2* (*UAS-drprRNAi*) (#67034; BDSC) [43], and *P{TRiP.HMJ02128}attP40* (*UAS-sharkRNAi*) (#42555; BDSC). These *UAS-RNAi* stocks can efficiently deplete the relevant mRNAs by combining them with Gal4 drivers. *P{w[+mC]=UAS-GFP.dsRNA.R}142* (#9330; BDSC) was used as a control for the *RNAi* experiments. To induce ectopic expression of the genes, the *UAS* stocks *UAS-Dro* and *UAS-CecA1* were used (gifts from B. Lemaitre, École Polytechnique Fédérale de Lausanne, Lausanne, Switzerland [9,44]). For the down-regulation of the innate immune pathways, *imd^1^* and *Toll^1-rxa^* were used (gifts from Y. Yagi, Nagoya University, Nagoya, Japan [25]). *UAS-Annexin V-GFP* (a gift from C. Han, Cornell University, Ithaca, NY, USA [45]) was used to visualize phosphatidylserine on the outside of the plasma membrane.

All *Drosophila* stocks were maintained on standard cornmeal food, as previously described [46]. Per liter of water, 40 g of dried yeast (Asahi Group, Tokyo, Japan), 40 g of corn flour (Nippun, Tokyo, Japan), 100 g of glucose (Kato Chemical, Aichi, Japan), and 7.2 g of agar powder (Matsuki Agar, Nagano, Japan) were added, and 5 mL of a 10% methyl para hydroxybenzoate solution and 5 mL of propionic acid (Tokyo Kasei Kogyo, Tokyo, Japan) were added to 1 L of the fly food. Induction of Gal4-dependent gene expression was performed at 28 °C. Other experiments and stock maintenance were conducted at 25 °C.

### 2.2. Germline Transformation

The pUAST-CecA-CFLAGHA plasmid (UFO09387; DGRC Stock 1643259) that permits expression for Cecropin A fused with FLAG- and HA- tags at its carboxyl-terminal under the UAS sequences (BDGF Tagged ORF collection, Drosophila Genomics Resource Center (Bloomington, IN, USA)). The plasmid DNA was injected into *Drosophila* embryos via PhiC31 integrase-mediated germ line transformation (BestGene Inc. (Chino Hills, CA, USA)).

### 2.3. Visualization of AMP Gene Expression in Drosophila Larvae Using Green Fluorescent Protein (GFP) Reporters

Mature third-instar larvae carrying the GFP reporters that monitored *Dro* and *CecA1* gene expression were collected and fixed on double-sided tape. The larvae were observed using an SZX7 stereo fluorescence microscope (OLYMPUS, Tokyo, Japan) equipped with a DIGITAL SIGHT DS-Fi2 digital camera (Nikon, Tokyo, Japan). Bright-field images and fluorescent images were acquired using a DS-L3 camera control unit (Nikon, Tokyo, Japan). Pairs of FB from mature third instar larvae were observed using an SZX7 stereo fluorescence microscope (OLYMPUS, Tokyo, Japan) equipped with a DIGITAL SIGHT DS-Fi2 digital camera (Nikon, Tokyo, Japan) and a DS-L3 camera control unit (Nikon, Tokyo, Japan) to obtain bright-field and fluorescence images.

### 2.4. Preparation of Fixed Samples to Measure LG Size

The *mxc^mbn1^* was maintained heterozygous for the *FM7a*, *P{w[+mC]=sChFP}1* balancer. In the stock, the larvae that did not express RFP were selected as the *mxc^mbn1^* hemizygotes [25]. The larval LGs are attached along the dorsal vessel and have three lobe-like structures, which are paired on the left and right sides [47]. The first lobe contains mature hemocytes in the cortical zone, and undifferentiated hematopoietic precursors in the medullary zone [48,49]. Larval LGs collected from mature third-instar larvae were fixed in 4% paraformaldehyde for 15 m. DNA was stained with a 4′,6-diamidino-2-phenylindole (DAPI) solution (1 µg/mL in PBS (Wako Pure Chemicals, Osaka, Japan)). The fixed LG specimen was mounted in a mounting medium (Vector Laboratories, Newark, CA, USA) under a cover glass, and mildly flattened under constant pressure using an apparatus so that the tissue spread out into thin cell layers with a constant thickness [25]. To quantify the size of each DAPI-stained LG, the entire area of each hemisphere of the LG was measured on the acquired fluorescence image using Image J software ver. 1.54p (https://imagej.nih.gov/ij/ accessed on 31 January 2025).

### 2.5. LG Immunostaining

To observe apoptosis in the LGs, the fixed LG samples were incubated with an anti-cDcp1 (Asp215) antibody (1:500; #9578, Cell Signaling Technology, Danvers, MA, USA), which recognizes cells undergoing apoptosis in *Drosophila* [25,50,51]. To visualize proliferating cells in the LGs, an anti-PH3(Ser10) antibody (1:1000; #06-570, Merk-Millipore, Burlington, MA, USA), which recognizes mitotic cells [25,50,52], was used. After repeated washing, the secondary antibody conjugated with Alexa Fluor 488 (1:400; #A11008, Molecular Probes, Eugene, OR, USA) was added to detect the primary antibodies. DAPI was used for DNA staining. The stained LG samples mounted in VECTASHIELD Mounting Medium (Vector Laboratories, Newark, CA, USA) were observed with an inverted fluorescence microscope IX81 (OLYMPUS, Tokyo, Japan) equipped with an ORCA-R2 digital CCD camera (Hamamatsu Photonics, Shizuoka, Japan). Fluorescence images were acquired using MetaMorph^®^ 7.6 Software (Molecular Devices, San Jose, CA, USA) and processed in Adobe Photoshop CS (Adobe, San Francisco, CA, USA). The areas emitting the immunofluorescent signals in each LG hemisphere were measured on the fluorescence images with Image J. The percentage of the fluorescent area in the anterior LG lobe was calculated.

### 2.6. Detection of Phosphatidylserine (PS) Exposed on Cell Membrane Surface in LG

To detect PS on the surface of the LG cells, GFP-tagged Annexin V—which binds to PS with a high affinity—was expressed in the FB using the *r4-Gal4* driver. When we observed PS on the LGs of the larvae with the precursor cell-specific depletion of *xkr* mRNA using the *upd3-Gal4* driver, we could not induce Annexin V-GFP expression via the simultaneous use of *r4-Gal4*. Thus, larval LGs collected from mature third-instar larvae were incubated with 5% Alexa Fluor 594-conjugated Annexin V (#A13203, Life Technologies, Foster City, CA, USA) for 30 m. The LGs were fixed in 4% paraformaldehyde for 15 m. DAPI staining and fluorescence imaging were then carried out as described above. The GFP fluorescence-positive areas in the anterior lobes of each LG hemisphere on the fluorescence images were measured using Image J. The percentage of the fluorescent area in the whole LG lobe region was calculated.

### 2.7. Immunostaining of Circulating Hemocytes

A single larva at the third instar stage was transferred into *Drosophila* Ringer’s solution (DR) on a glass slide. Subsequently, only the larval epidermis was cut using a set of fine forceps to allow for the release of the circulating hemocytes into the DR outside the larva. After an aliquot of the DR containing circulating hemocytes was placed on the glass slide, the hemocytes were fixed in 4% paraformaldehyde for 10 m. Immunostaining of the hemocytes was performed using anti-HA-tag rabbit IgG (1:1000; #3724, Cell Signaling Technology, Danvers, MA, USA), as described above. The fluorescence intensity of the hemocytes was quantified using Image J.

### 2.8. Microinjection of Synthetic Cecropin A Peptides

A 50 μM solution of synthetic cecropin A (#C6830, Sigma-Aldrich, St. Louis, MO, USA) dissolved in PBS containing a red food color was prepared for microinjection [53]. Using red pigment as an injection marker and a glass needle, approximately 0.1 µL of the AMP solution was injected into the posterior–ventral area of the recipient larva at the third instar, using glass needles. The needles were prepared from G1.2 capillaries (outer diameter of 1 mm, Narishige Co., Tokyo, Japan) using a glass puller (PN-31, Narishige Co., Tokyo, Japan). They were ground against the side of the microscope glass to sharpen the tip. After injection, the larvae were placed on a piece of wet filter paper for 1 h to recover from the damage and provided with standard food overnight before observation.

### 2.9. Quantitative Reverse Transcription-PCR (qRT-PCR) Analysis

Total RNA was extracted from 14 to 18 pairs of FB of mature third-instar larvae using Trizol Regent^®^ (Invitrogen, Waltham, MA, USA). After treatment with DNase I (Epicentre Technologies, Madison, WI, USA) to remove mixed genomic DNA, the purity of the RNA was checked by ensuring that the A260–A280 ratio of each RNA sample was between 1.8 and 2.0. cDNA was synthesized from the total RNA using a PrimeScript High-Fidelity RT-PCR Kit (TaKaRa, Clontech Laboratories, Shiga, Japan). Real-time PCR reactions were performed on a Thermal Cycler Dice^®^ Real-Time System III (TaKaRa Bio, Shiga, Japan) using TB Green^®^ Premix Ex Taq™ II Tli RNaseH Plus (TaKaRa Bio, Shiga, Japan). The PCR reaction was carried out using a cycling program consisting of initial denaturation at 95 °C for 5 m, followed by 40 cycles at 95 °C for 5 s and 60 °C for 30 s. The temperature was increased from 60 °C to 95 °C at a rate of 0.1 °C/s. Real-time PCR was performed using a Thermal Cycler Dice^®^ Real-Time System III (TaKaRa Bio., Shiga, Japan), using TB Green Premix Ex Taq II (#RR820A, TaKaRa Bio, Shiga, Japan). Each sample was analyzed in triplicate on a PCR plate, and the final results were obtained by averaging three biological replicates. For quantification, the ∆∆Ct method was used to determine the differences between target gene expression and that of the reference gene, Rp49. Three identical PCR reaction reagents were prepared for one cDNA sample, and the mean and standard deviation of the mRNA amounts were calculated. The mRNA amounts were analyzed using the ΔΔCt method. The following primer sequences were used in the real-time quantitative PCR: RP49-Fw, 5′-TTCCTGGTGCACAACGTG-3′, and RP49-Rv, 5′-TCTCCTTGCGCTTCTTGG-3′; Cecropin A1-Fw, 5′-TCTTCGTTTTCGTCGCTCTC-3′, and Cecropin A1-Rv, 5′-CTTGTTGAGCGATTCCCAGT-3′; and Drosocin-Fw, 5′-TCAGTTCGATTTGTCCACCA-3′, and Drosocin-Rv, 5′-GATGGCAGCTTGAGTCAGGT-3′.

### 2.10. Statistical Analysis

Welch’s *t*-test and one-way ANOVA for multiple comparisons were used to assess statistical differences. Unless otherwise stated, a one-way ANOVA for multiple comparison with Bonferroni correction was used for statistical comparisons. Sample sizes and *p*-values are provided in the figure legends. A *p*-value of 0.05 or less was considered statistically significant. The results of each tabulation are displayed as scatter plots or bar charts, which were created using GraphPad Prism 6 (GraphPad Software, Boston, MA, USA).

## 3. Results

### 3.1. Induction of AMP Genes Encoding Drosocin and Cecropin A in the FB of mxc^mbn1^ Mutant Larvae

The transcription of genes encoding the five major AMPs was induced in the FB of the *mxc^mbn1^* mutant larvae [25]. Thus, we first examined whether the other two major AMP genes encoding Drosocin (*Dro*) and Cecropin A (*CecA1*) showed a consistent induction of transcription in the mutant FB. First, we visualized *Dro* gene expression in the FB of normal (*w/Y*; *Dro-GFP/+*) and *mxc^mbn1^* mutant larvae (*mxc^mbn1^/Y*; *Dro-GFP/+*) at the third instar stage using the green fluorescent protein (GFP) reporter. GFP fluorescence was not detected in the normal control (Figure 1a′). In contrast, 35% of the *mxc^mbn1^* larvae showed GFP fluorescence in the FB (Figure 1b′), although the fluorescence intensity was weaker than that observed in the case of bacterial infection. Similarly, *CecA1* gene expression in the FB of control (*w/Y*; *CecA1-GFP/+*) and *mxc^mbn1^* mutant (*mxc^mbn1^/Y*; *CecA1-GFP/+*) larvae was visualized using the GFP reporter. GFP fluorescence was not detected in the controls (Figure 1c′) but was detected in the FB of 33% of the mutant larvae (Figure 1d′). These findings suggest that the *Dro* and *CecA1* genes were upregulated in the FB of *mxc^mbn1^* larvae.

To confirm the upregulation of *Dro* and *CecA1* in the *mxc^mbn1^* larvae, we performed quantitative reverse transcription-PCR (qRT-PCR) experiments using total RNA from the FB of the normal (*w/Y*) and mutant (*mxc^mbn1^/Y*) larvae. The average mRNA levels of *Dro* and *CecA1* increased approximately 50-fold and 7-fold, respectively, in the mutants compared to the normal controls (Figure 1f). Thus, the *Dro* and the *CecA1* genes were overexpressed in the FB of *mxc^mbn1^* larvae harboring LG tumors.

### 3.2. Declines in Dro and CecA1 mRNA Levels and Enhancement of LG Hyperplasia in mxc^mbn1^ Larvae via Half-Dose Reduction in the Genes Encoding the Innate Immune Pathway Factors

Using genetic analysis, we investigated whether *Dro* and *CecA1* were induced in the *mxc^mbn1^* larvae via the activation of innate immune pathways. The *mxc^mbn1^* mutants heterozygous for *Toll* or *imd* mutations exhibited reduced mRNA levels of the five other AMP genes [25]. In *mxc^mbn1^* mutants heterozygous for a loss-of-function mutation for *Toll* (*mxc^mbn1^/Y*; *Toll^1-RXA^/+*) or *imd* (*mxc^mbn1^/Y*; *imd^1^/+*), the mRNA levels of *Dro* and *CecA1* in the FB were quantified using qRT-PCR. The *Dro* mRNA level declined to approximately 7% of that of *mxc^mbn1^* (*mxc^mbn1^/Y*) in the mutant larvae heterozygous for the *Toll* mutation and to approximately 3% in *imd* heterozygous mutants (Figure 2a). Consistently, the *CecA1* mRNA levels declined to 35% of those of *mxc^mbn1^* in the mutant larvae heterozygous for the *Toll* mutation and by approximately one-third in *imd* heterozygous mutants (Figure 2b).

Next, we observed whether the growth of the LG tumors was enhanced when the mRNAs of AMP genes were downregulated in *mxc^mbn1^* larvae heterozygous for the *Toll* or *imd* mutation (Figure 2c–f). The size of the entire LG lobe region (mean: 0.55 mm^2^) increased by 1.13-fold in *mxc^mbn1^* larvae heterozygous for the *Toll* mutation compared to that in *mxc^mbn1^* larvae without the mutation (mean: 0.48 mm^2^) (Figure 2g). Consistently, the LG tumors in *mxc^mbn1^* larvae heterozygous for the *imd* mutation (mean: 0.67 mm^2^) were 1.38-fold larger (Figure 2g) than those in *mxc^mbn1^* larvae (mean: 0.48 mm^2^). In summary, LG hyperplasia was enhanced in *mxc^mbn1^* mutants heterozygous for *Toll* or *imd* mutations. Thus, LG tumors are suppressed by the gene products of *Dro* and *CecA1*, which are induced via innate immune pathways.

### 3.3. LG Hyperplasia Suppression in mxc^mbn1^ Larvae via Overexpression of Dro or CecA1 Gene in the FB

Further, we investigated the anti-tumor potential of Drosocin and Cecropin A. The FB-specific overexpression of either gene (*w/Y*; *r4>Dro* or *w/Y*; *r4>CecA1*) did not affect LG size relative to that of the normal control larvae (*w/Y*; *r4>+*) (Figure 3a–c). We next compared the LG size of *Dro*-overexpressing *mxc^mbn1^* larvae in the FB (*mxc^mbn1^/Y*; *r4>Dro*) with that of *mxc^mbn1^* larvae (*mxc^mbn1^/Y*; *r4>+*) (Figure 3d,e). The average size of the entire LG lobe region (mean: 0.18 mm^2^) was reduced to approximately one-third of that of the *mxc^mbn1^* larvae (mean: 0.48 mm^2^) (Figure 3g). These results indicate the potential anti-tumor effect of Drosocin on LG tumors in *mxc^mbn1^* mutants. Next, we compared the LG size of *mxc^mbn1^* larvae with FB-specific *CecA1* overexpression (*mxc^mbn1^*/*Y*; *r4>CecA1*) to that of *mxc^mbn1^* larvae (*mxc^mbn1^/Y*; *r4>+*) (Figure 3d,f). The LG size (mean: 0.15 mm^2^) was reduced to approximately one-third of that of the control larvae (*mxc^mbn1^/Y*; *r4>+*) (mean: 0.48 mm^2^) (Figure 3g). These results indicate the anti-tumor potential of Drosocin and Cecropin A, which both suppressed LG tumor growth in *mxc^mbn1^* larvae.

### 3.4. Dro or CecA1 Overexpression in FB Enhanced Apoptosis in the LG Tumors of mxc^mbn1^ Larvae

To elucidate the mechanism underlying the anti-tumor effect of Drosocin and Cecropin A, we investigated whether these two AMPs also induced apoptosis in LG tumors, similarly to the other five AMPs [25]. First, we confirmed that the FB-specific overexpression of *Dro* (*w/Y*; *r4>Dro*) or *CecA1* (*w/Y*; *r4>CecA1*) did not influence LG size in normal larvae (Figure 3a–c). No anti-cDcp1 immunostaining signals were found in the total lobe regions of the LGs of the normal larvae (*w/Y*; *r4>+*) or the control larvae with FB-specific overexpression of *Dro* (*w/Y*; *r4>Dro*) and *CecA1* (*w/Y*; *r4>CecA1*) (Figure 4a′–c′). By contrast, the LGs in *mxc^mbn1^* mutant larvae with the FB-specific overexpression of *Dro* (*mxc^mbn1^/Y*; *r4>Dro*) showed signals corresponding to apoptotic cells in an average of 22.1% of the total LG lobe area, a 1.8-fold higher value than the average of 12.9% observed in *mxc^mbn1^* larvae (*mxc^mbn1^/Y*; *r4>+*) (Figure 4d′,e′,g). These results suggest that Drosocin induces apoptosis in *mxc^mbn1^* LG tumors. Consistently, the FB-specific overexpression of *CecA1* in *mxc^mbn1^* larvae (*mxc^mbn1^/Y*; *r4>CecA1*) also increased the apoptosis area to an average of 29.4% of the total lobe area of LGs in larvae at the third instar stage. This percentage was 2.3-fold higher than the average of 12.9% in the *mxc^mbn1^/Y*; *r4>+* larvae (Figure 4d′,f′,g). These results suggest that Cecropin A could induce apoptosis in *mxc^mbn1^* mutant LG tumors.

To further confirm whether apoptosis induction occurs in a tumor-specific manner, we overexpressed *Dro* or *CecA1* specifically in the FB of normal larvae and investigated apoptosis in larval tissues such as imaginal discs using anti-cDcp1 immunostaining. Few signals were observed in the wing imaginal discs, similar to the controls (Appendix A). Thus, Drosocin and Cecropin A-induced apoptosis specifically in the LG tumors of the *mxc^mbn1^* larvae.

### 3.5. Dro or CecA1 Knockdown in the FB Enhanced LG Hyperplasia and Suppressed Apoptosis in LG Tumors in mxc^mbn1^ Larvae

Conversely, we investigated whether the knockdown of these two AMPs influenced the growth of LG tumors. We depleted *Dro* mRNA (*w/Y*; *r4>DroRNAi*) or induced dsRNA against *GFP* mRNA in the FB of normal controls (*w/Y*; *r4>GFPRNAi*) (Figure 5a,b). The average LG tumor sizes of these controls were 0.032 mm^2^ and 0.031 mm^2^, respectively (Figure 5g). No significant differences were observed. In contrast, the average size of the LG lobe area in mutant larvae with the *Dro* depletion (*mxc^mbn1^/Y*; *r4>DroRNAi*) (mean: 0.51 mm^2^) was 1.43 times larger than that in the mutant larvae expressing non-specific dsRNA against GFP mRNA (mean: 0.36 mm^2^) (*mxc^mbn1^/Y*; *r4>GFPRNAi*) (Figure 5g). These results indicated the anti-tumor effect of Drosocin, which suppressed LG tumor growth in the *mxc^mbn1^* larvae.

Consistently, the area of the entire LG lobe region in the mutant larvae with *CecA1* depletion (mean: 0.48 mm^2^) (*mxc^mbn1^/Y*; *r4>CecA1RNAi*) was 1.34 times larger than that of the depletion control larvae (*mxc^mbn1^/Y*; *r4>GFPRNAi*) (mean: 0.36 mm^2^) (Figure 5g). These results indicate that Cecropin A also has an anti-tumor effect on *mxc^mbn1^* LG tumors.

Next, we investigated whether their depletion influenced the LG tumors. First, we confirmed that neither the depletion of *Dro* mRNA (*w/Y*; *r4>DroRNAi*) nor the ectopic expression of non-specific dsRNA against *GFP* mRNA (*w/Y*; *r4>GFPRNAi*) in the FB influenced the apoptosis area stained with the anti-cDcp1 antibody in the LGs of normal larvae (Figure 6a′,b′). In *mxc^mbn1^* larvae with the FB-specific expression of dsRNA against *GFP* mRNA, we observed apoptosis signals in an average of 17.7% of the total LG lobe area (*mxc^mbn1^/Y*; *r4>GFPRNAi*). In contrast, in the LGs of the mutant larvae with the FB-specific depletion of *Dro* (*mxc^mbn1^/Y*; *r4>DroRNAi*), we observed apoptosis signals in an average of 6.2% of the total lobe areas (Figure 6d′,e′). This percentage declined to 35% of that in the LGs of the mutant larvae with non-specific dsRNA expression (Figure 6g). These results indicate that Drosocin induces apoptosis in LG tumors in *mxc^mbn1^* mutant larvae.

Next, we confirmed that neither the depletion of *CecA1* mRNA nor the ectopic expression of non-specific dsRNA against *GFP* mRNA affected the FB-influenced apoptosis areas in the LGs of normal larvae (Figure 6a′,c′). In contrast to the average percentage of 17.7% in the mutant LGs (*mxc^mbn1^/Y*; *r4>GFPRNAi*), apoptosis signals were observed in an average of 9.3% of the total LG lobe area in the mutant larvae (Figure 6d′,f′). This percentage declined by approximately 50% in the LGs of mutant larvae with non-specific dsRNA expression (Figure 6g). These results indicate that Cecropin A induced apoptosis in the LG tumors of the *mxc^mbn1^* larvae.

Furthermore, we investigated whether synthetic cecropin A peptides injected into the *mxc* mutant larvae could also induce apoptosis in the LG tumors. We injected a solution of synthetic cecropin A of *H. cecropia* into the *mxc^mbn1^* larvae and observed whether the areas of apoptosis in the mutant LGs increased compared to the areas of apoptosis in the mutant larvae injected with PBS only (Figure 7a–d). The average percentage of the cDcp1-positive areas within the mutant LGs increased by 48.3% compared with the PBS-injected control (Figure 7e). These results suggest that synthetic cecropin A peptides of a different species can also induce apoptosis in *Drosophila* LG tumors.

### 3.6. Dro or CecA1 Overexpression in the FB Did Not Affect Cell Proliferation in the LGs of mxc^mbn1^ Larvae

Another potential anti-tumor mechanism of these two AMPs is the inhibition of LG cell proliferation. To test this possibility, we observed mitotic cells in the LGs of mutant larvae overexpressing *Dro* or *CecA1* in the FB. Anti-phosphorylated histone H3 immunostaining revealed mitotic cells at similar low frequencies in the total lobe regions of LGs in normal larvae (*w/Y*; *r4>+*) and in those of control larvae with the FB-specific overexpression of Dro (*w/Y*; *r4>Dro*) and CecA1(*w/Y*; *r4>CecA1*) (Appendix A). Next, we observed mitotic cells in 14.8% of the total lobe area in the *Dro*-overexpressing mutant larvae (*mxc^mbn1^/Y*; *r4>Dro*) LG (Appendix A). The difference was not statistically significant (Appendix A), although this percentage was slightly reduced from the average (16.0%) observed in the LGs of *mxc^mbn1^/Y*; *r4>+* larvae. Thus, Drosocin overexpression failed to alter cell proliferation in the LG tumors of the *mxc^mbn1^* mutants.

We also induced *CecA1* overexpression in the FB and performed anti-PH3 immunostaining of LGs in the larvae. Consistently, inducing *CecA1* expression in the FB of *mxc^mbn1^* mutant larvae (*mxc^mbn1^/Y*; *r4>CecA1*) resulted in immunostaining of an average of 16.4% of the total lobe area in the LGs of the larvae, similar to the 16.0% observed in the *mxc^mbn1^* larvae (*mxc^mbn1^/Y*; *r4>+*) (Appendix A). This difference was not statistically significant (Appendix A). Thus, similar to Drosocin, Cecropin A overexpression does not affect cell proliferation in *mxc^mbn1^* LG tumors.

### 3.7. Incorporation of Cecropin A into Circulating Hemocytes in Tumor-Bearing mxc^mbn1^ but Not in Control Larvae

Next, we investigated how Cecropin A, when synthesized in the FB and secreted into the hemolymph, acts on LG tumors (Figure 8). Other AMPs—Drosomycin, Diptericin, and Defensin—are incorporated into hemocytes in the hemolymph of tumor-bearing larvae in *mxc^mbn1^*. We therefore investigated whether Cecropin A has a similar property. HA-tagged Cecropin A was expressed in the FB of *mxc^mbn1^* larvae (*mxc^mbn1^/Y*; *r4>CecA-HA*), and immunostaining of the circulating hemocytes was performed with an anti-HA antibody to determine if it would be taken up by the cells. We observed immunostaining signals indicating the presence of Cecropin A in 59.5% of hemocytes in the *mxc* mutant larvae (Figure 8b″). In contrast, we observed few signals in the hemocytes of the control larvae (Figure 8a″).

### 3.8. Drosomycin Incorporation into Hemocytes Required Draper Signaling in mxc^mbn1^ Larvae

Next, we investigated the specific mechanism by which AMPs known to possess anti-tumor properties are taken up into circulating hemocytes in the *mxc* mutant larvae bearing LG tumors. As we speculated that endocytosis factors mediate this process, we depleted mRNA for Draper, a phagocytosis factor, in the hemocytes. Since antibodies against Cecropin A or other well-known AMPs were unavailable, we used the stock expressing GFP-tagged Drosomycin under its gene promoter (*Drs::GFP*). This AMP was also incorporated into the circulating hemocytes in the mutant larvae [25]. We examined the cellular localization of this AMP in the hemocytes using this GFP fusion protein. GFP signals indicative of Drosomycin were not detected in the hemocytes of control larvae (*w*, *Drs::GFP/Y*) (Figure 9a″). By contrast, a 10-fold increase in fluorescence intensity was detected in the cytoplasm of hemocytes in the *mxc^mbn1^*, *Drs::GFP/Y* larvae (Figure 9b″). There were no alterations in the number of circulating hemocytes in control larvae with the hemocyte-specific depletion of phagocytosis factors, Draper and Shark (*w/Y*; *He>drprRNAi* and *w/Y*; *He>sharkRNAi*), compared to the normal larvae (*w/Y*), suggesting that the depletion of these factors did not affect hemocyte survival. When we next depleted these mRNAs specifically in the hemocytes of the mutant larvae (Figure 9c,d), the frequency of GFP^+^ cells decreased to 5.1% and 3.5%, respectively, compared to 39.5% of hemocytes without the depletion (*mxc^mbn1^*, *Drs::GFP*; *He>+*) (Figure 9e). The intensities of the GFP signal indicative of Drosomycin within the hemocytes also decreased below a background level (Figure 9b″–d″). From these results, we concluded that the uptake of *Drs* is inhibited when *drpr* or *shark* mRNA is depleted in hemocytes. In other words, Draper signaling is indispensable for the uptake of Drosomycin into hemocytes in *mxc^mbn1^* larvae.

Furthermore, to exclude the possibility that gene transcription was induced in the mutant hemocytes, we monitored its gene expression using a *Drs-YFP* reporter. We did not see any YFP signals indicating its expression in the *mxc^mbn1^* mutants and the controls (Appendix A). We concluded that *Drs* was not transcribed in circulating hemocytes in the *mxc^mbn1^* mutant larvae. Based on these results, Drosomycin was incorporated, but not transcribed, into circulating hemocytes, specifically those of the *mxc^mbn1^* larvae bearing the LG tumors, which require the phagocytosis factors Draper and Shark.

### 3.9. Phosphatidylserine Localization on the Plasma Membrane Surface in LG Tumors

Phosphatidylserine (PS) is exposed on the lipid bilayer surface in cancer cells and serves as a marker for phagocytosis by macrophages. In the lipid bilayers of tumors arising on the wing discs in *Drosophila dlg* mutants, more PS is exposed on the surface than in the wild type [28]. Therefore, we hypothesized PS localization on the cell surface in the LG tumors of *mxc^mbn1^* mutants. To test this, we induced the ectopic expression of Annexin V-GFP, which has a strong affinity for PS, in the FB to allow for hemolymph secretion. We observed GFP fluorescence in the LG tumors of the *mxc^mbn1^* larvae (*mxc^mbn1^/Y*; *r4>Annexin V-GFP*) but not the normal controls (*w/Y*; *r4>Annexin V-GFP*) (Figure 10a″,b″). The Annexin V-binding region (mean: 26.8%) increased in the LGs of the *mxc^mbn1^* mutants compared to the normal controls (Figure 10c). Thus, we concluded that PS was exposed on the cell membrane surface in the *mxc^mbn1^* LG tumors.

### 3.10. Xkr Scramblase Knockdown Canceled PS Localization on Surface of LG Cells and Led to Enhancement of LG Hyperplasia in mxc^mbn1^

To confirm whether PS is indeed used as a target of tumor suppression, we depleted the mRNA required for scramblase to expose PS to the cell surface and examined its effect on LG tumor growth. For this purpose, we depleted *xkr scramblase* mRNA using the known *UAS-xkrRNAi* stock [54] and the *upd3-Gal4* driver, in which Gal4 is expressed in the undifferentiated cell region of the LG, the origin from which LG tumors arise [36]. The fluorescence indicating Annexin V binding almost disappeared from the *xkrRNAi* LGs, as shown in Figure 11c. In other words, the PS signal exposed to the cell surface in the LG tumor region was reduced. In these larvae, the LG tumor size increased to an average of approximately twice that of the control (Figure 11d). Based on these results, we conclude that PS exposure on the surface of tumor cells is required to target anti-tumor proteins, such as Drosomycin, to the LG tumors.

## 4. Discussion

### 4.1. Cecropin A and Drosocin Induction via Innate Immune Pathways in the FB of mxc^mbn1^ Mutant Larvae Bearing the Tumors

This study demonstrated that the levels of mRNAs encoding Cecropin A and Drosocin were elevated in the FB of *mxc^mbn1^* larvae bearing LG tumors. This upregulation depended on both the Toll- and Imd-mediated immune pathways. These findings suggest that the innate immune system plays a role in suppressing cancer cells, even in *Drosophila*. In mammals, both the innate and acquired immune systems are involved in eliminating cancer cells [55,56,57,58]. In contrast, invertebrates lack an acquired immune system. Consequently, the tumor-suppressive effects described in this study are attributable to the innate immune system. This process involves the participation of plasmatocytes, which are macrophage-like cells in *Drosophila* hemolymph [31,35]. In *mxc^mbn1^* larvae with LG tumors, circulating hemocytes can recognize damage to the basement membrane and subsequently accumulate on the tumors [34]. Additionally, as the LG tumor expresses Eiger, a tumor necrosis factor (TNF) orthologue, the hemocytes recognize it via Eiger receptors on the cell surface, thereby inducing Turandot family proteins with anti-cancer effects [31]. Therefore, in *Drosophila*, cells of the innate immune system can recognize the tumor and transmit this information to the FB. The induction of Cecropin A and Drosocin is interpreted to be induced via a similar inter-tissue communication. Furthermore, crosstalk between the FB and hemocytes is also needed to suppress tumors that arise in the wing imaginal discs. In this instance, active Spätzle, generated by reactive oxygen species accumulated in hemocytes, activates Toll in the FB cells [27,34]. In contrast, the mechanism by which the Imd pathway is activated in response to tumors remains unelucidated. The present study demonstrates that the induction of Cecropin A and Drosocin in the FB is diminished in *mxc^mbn1^* mutants when the doses of the genes for signaling factors in the Imd pathway are halved. The expression of these two AMPs during bacterial infection is regulated by both the Toll and Imd pathways [9]. It was reported that these two pathways engage in cross-talk [59]. Consequently, a Toll-mediated signal is triggered by Spätzle around tumor-responsive hemocytes associated with the FB. This may also activate the Imd pathway, eventually inducing target AMP gene expression.

### 4.2. Cytotoxic Effects of Cecropin A and Drosocin on Tumors in Drosophila Larvae

This study showed that Cecropin A and Drosocin overexpression in the FB increased apoptosis induction and consequently reduced LG tumor size in *mxc^mbn1^* larvae. Therefore, we conclude that these two AMPs have anti-tumor effects against LG tumors in *mxc^mbn1^* larvae. These results are consistent with previous findings that five other AMPs—Drosomycin, Diptericin, Defensin, Metchinikowin, and Attacin A—and two Turandot family proteins—TotB and TotF—possess anti-tumor properties that suppress tumor growth in *Drosophila* [25,28,50]. Moreover, this study demonstrated that synthetic cecropin A peptides of the Cecropia moth can also stimulate apoptosis in LG tumors in *Drosophila*. Consistently, housefly cecropin also induces apoptosis in human hepatocellular carcinoma cells without affecting normal cells [60]. Synthetic cecropin A demonstrates anti-cancer effects against leukemia cell lines [61,62]. These studies were performed to determine the anti-cancer properties of AMPs of other species in vitro. By contrast, we demonstrated that cecropin A stimulated apoptosis in tumors in living organisms. Our experimental system may be used as a simple in vivo model to determine if anti-cancer drug candidates stimulate apoptosis and suppress tumor growth. Cecropin B also exhibits selective anti-tumor activity against human cancer cells, which can be applied to anti-cancer cell therapy [63,64]. Therefore, cecropin B may exhibit a stronger anti-tumor effect. This will be investigated in a future study.

### 4.3. The Tumor-Specific Effect of Cecropin A and Drosocin on LG Tumors in mxc^mbn1^ Larvae May Be Determined by the Circulating Hemocytes That Take Up AMPs

A detailed mechanism of the cytotoxic effects that AMPs exert on tumors while sparing normal cells remains unclear [64,65]. Drosomycin, Defensin, and Turandots are taken up by circulating hemocytes in *mxc^mbn1^* mutant larvae-bearing tumors but not normal larvae [25,31,50]. Basement membrane damage in LG tumors is involved in tumor cell recognition by circulating hemocytes, resulting in their accumulation [34]. Eiger, a *Drosophila* TNF orthologue, is ectopically expressed in the LG tumors of *mxc^mbn1^* larvae. When hemocytes receive it via the Eiger receptor on the cell surface, Upd3, a *Drosophila* functional IL-6 orthologue, is induced. This is essential in transmitting information to the FB [31]. Circulating hemocytes that recognize LG tumors may take up the AMPs in this way and accumulate in LG tumors again. The AMPs are then released from the hemocytes and can act locally on the tumors.

This study demonstrated that the circulating hemocytes of *mxc^mbn1^* larvae take up Cecropin A, although the mechanism remains unclear. The localization of HA-tagged AMP specifically induced using a FB-specific Gal4 driver was investigated via anti-HA immunostaining. It is challenging to knock down the genes required for hemocyte uptake using another driver in the same larvae. Antibodies to detect either AMP were unavailable, despite our efforts to generate them, but we will continue to develop them and investigate the localization of Drosocin. Instead, we investigated the uptake of Drosomycin using the strain in which the GFP-tagged peptides can be induced under its promoter. We observed GFP fluorescence in the hemocytes of *mxc^mbn1^* mutants but not normal larvae, even though the gene was not transcribed in the hemocytes of the mutant larvae. This is consistent with the results that HA-tagged Drosomycin and other AMPs, including Cecropin A, are incorporated into the circulating hemocytes of the mutant larvae [26] and this study. Using the stock expressing GFP-tagged Drosomycin, we demonstrated that the phagocytosis factors, Draper and Shark [66], are required for hemocytes to uptake the AMPs. The AMPs are possibly taken up into vacuoles, such as phagosomes, formed during phagocytosis. Previous studies described that Drosomycin is enclosed in cytoplasmic vesicles and transported to the endosome system when it is released from the FB into the hemolymph in response to bacterial infection [67,68]. In response to tumor cells, these incorporated AMPs are likely transported to the plasma membrane via a recycling endosome pathway [69]. Subsequently, they may be released from the cells via exocytosis [70]. This hypothesis requires validation in the future.

### 4.4. Restrictive Anti-Cancer Effects of AMPs on LG Tumor Cells in Which PS Was Exposed on the Plasma Membrane Surface

This study showed that Cecropin A or Drosocin overexpression in the FB of *mxc^mbn1^* larvae stimulated apoptosis and suppressed LG tumor growth. Furthermore, the absence of apoptosis induction in the tissues of normal larvae overexpressing Drosocin or Cecropin A suggested that these AMPs act in a strict tumor-specific manner. The cell membrane surface of normal cells is positively charged, whereas that of malignant tumor cells is negatively charged. This is due to the superficial presence of PS on the external plasma membrane surface [71]. In normal cells, PS is retained in the inner leaflet of the plasma membrane by scramblase [72]. In cancer cells, however, PS is exposed on the cell surface due to reduced scramblase activity [73]. The relationship between the density of PS on the surface and cell sensitivity to AMP has been suspected in several cancer cell lines, suggesting that PS plays a critical role in anti-cancer activity [74]. The negative surface charge of cancer cell membranes is shared by bacterial cells [66,75]. Thus, AMPs may attack tumors through a mechanism similar to that of bacteria. This implies that AMPs with positively charged amino acid sequences may easily bind to negatively charged cancer cell membranes. Upon insertion into the cell membrane, they can kill cancer cells by rupturing the membrane [76]. PS is prevalent on the surface of the plasma membrane in *Drosophila* wing disc tumors, and Defensin acts by marking this PS on the surface of the tumors [28]. This study observed that Annexin V, which acts through the hemolymph, binds to the cell surface in LG tumors, sparing healthy tissue. Thus, conceivably, PS is similarly exposed on the surface of LG tumor cells in *mxc^mbn1^* larvae, which may facilitate tumor cell targeting by AMPs. Thus, AMPs act using PS as a landmark, resulting in apoptosis. Taken together, we can speculate that when hemocytes recognize LG tumors, they take up AMPs via phagocytosis and are recruited very close to the LG tumor, where the AMPs are released. In addition to the role of hemocytes in the limitation of AMPs’ range of action, their possible target, PS on the tumor cell surface, further restricts their anti-tumor effect. Notably, an important factor in developing anti-cancer drugs is the absence of side effects. If the tumor-specific anti-cancer effects of these AMPs are proven and their mechanism of action elucidated, they are expected to constitute anti-cancer drugs with minimal side effects.

Finally, we discuss the limitations of this study and several issues that remain to be addressed in the future. First, Cecropin A was incorporated into hemocytes in larvae with tumors, while Drosocin uptake was not investigated. Antibodies that could be used for immunostaining were not obtained. This is a limitation of this study. In future studies, we propose establishing a UAS line that expresses HA-tagged Drosocin to confirm its tumor-dependent uptake. Second, although we found that synthetic AMP peptides of insects other than *Drosophila* were also able to induce apoptosis in the *mxc^mbn1^* tumors, many AMPs are not conserved between species. Exceptionally, some defensin family members are also conserved in mammals [77]. Mammalian synthetic defensins’ effects on LG tumors are another issue to be investigated. Some AMPs require post-translational modifications, such as glycosylation, to achieve their cytotoxic effects. If so, these effects may not be observed in synthetic peptides, representing a problem with this method. While these AMPs’ effects cannot be studied as readily as those of synthetic peptides, it is possible to determine whether tumor growth can be suppressed via the expression of mammalian AMPs in *Drosophila*. Finally, we showed that Drosomycin uptake requires the phagocytosis factors, Draper and Shark; however, how AMPs are taken into the cell remains to be determined. Previous studies suggest that they are inserted into the plasma membrane and that the cellular membrane is subsequently disrupted. It is not understood why such peptides do not act on the isolating membrane when they are enclosed in the phagosome. The mechanism by which the enclosed AMPs are secreted outside hemocytes must also be determined. We presume that exocytosis is involved, but it is necessary to verify this by conducting knockdown experiments with the genes required.

## 5. Conclusions

Two major AMPs in *Drosophila*, Cecropin A and Drosocin, are induced in *mxc* mutant larvae harboring LG tumors depending on the innate immune pathway. These AMPs specifically exert cytotoxic effects on the tumors by enhancing apoptosis. The AMPs synthesized in the FB are incorporated into macrophage-like blood cells in the mutant larvae, but not in normal larvae. Another AMP, Drosomycin, is incorporated via Draper-mediated phagocytotic signaling. After being transported to the vicinity of the tumors, the AMPs released from the blood cells may target phosphatidylserine exposed on the tumor surface for apoptosis induction. Synthetic cecropin A peptides of another organism also showed an apoptosis-inducing property. They are anticipated to form the basis for the development of anti-cancer drugs with minimal side effects.

## Figures and Tables

**Figure 1 cells-14-00389-f001:**
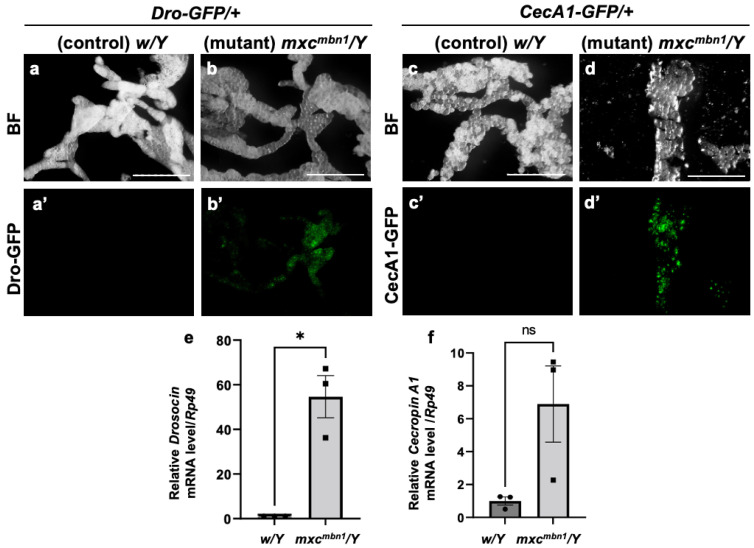
The expression of *Dro-GFP* and *CecA1-GFP* reporters in the fat body (FB) of *mxc^mbn1^* mutant larvae: (**a**,**b**) Bright-field (BF) stereomicroscopic images of the FB of mature third-instar larvae carrying the *Drosocin* (*Dro*)-*GFP* reporter. Scale bars: 500 µm. (**a′**,**b′**) Green fluorescent protein (GFP) fluorescence images of the FB of mature third-instar larvae with the *Dro*-*GFP* reporter. (**c**,**d**) BF stereomicroscopic images of the FB in a mature third-instar larva carrying the *Cecropin A1* (*CecA1*)-*GFP* reporter. (**c′**,**d′**) GFP fluorescence images of the FB of the larvae with the *CecA1*-*GFP* reporter. (**a**,**c**) Normal control (*w/Y*) and (**b**,**d**) *mxc^mbn1^* mutant (*mxc^mbn1^/Y*) larvae. (**e**,**f**) mRNA quantification of *Dro* and *CecA1* using quantitative reverse transcription-PCR (qRT-PCR). The X-axis of each graph shows the mRNA levels of the normal control (*w/Y*) and *mxc^mbn1^* (*mxc^mbn1^/Y*) larvae from left to right; the Y-axis shows the mRNA levels of the target gene relative to the endogenous control gene (*Rp49*). (**e**,**f**) mRNA levels of the *Dro* (**e**) and *CecA1* (**f**) genes. Significant differences between the experimental groups were determined using Welch′s *t*-test (* *p* < 0.05, ns: not significant). The error bars indicate the standard error of the mean (SEM).

**Figure 2 cells-14-00389-f002:**
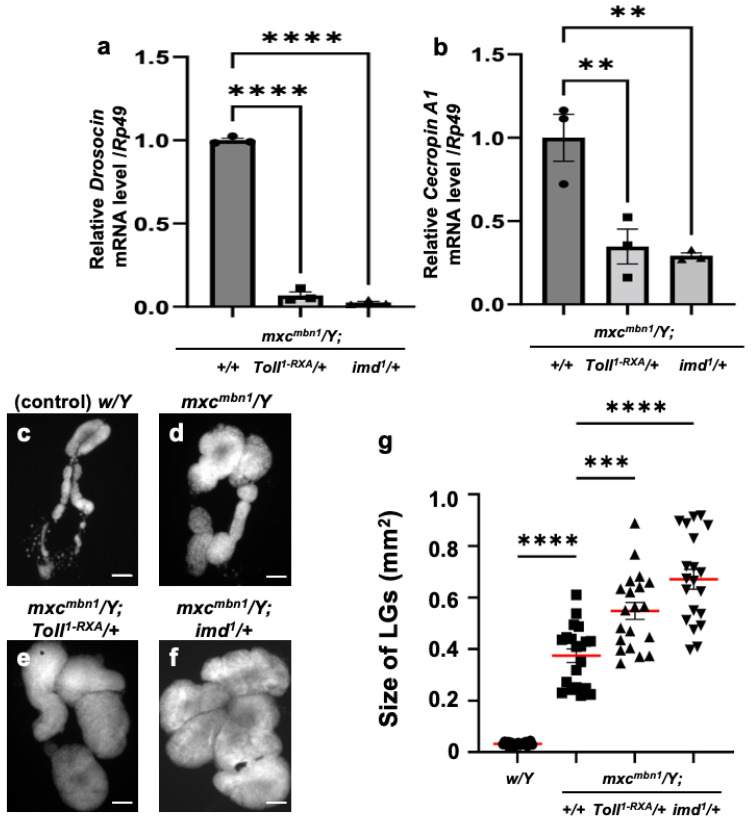
The mRNA levels of *Dro* and *CecA* genes in the fat body and the LG tumor size of *mxc^mbn1^* larvae heterozygous for mutations of the genes encoding the factors in innate immune pathways: (**a**,**b**) Quantification of mRNA levels of the *Dro* gene encoding Drosocin and the *CecA1* gene encoding Cecropin A using qRT-PCR. X-axis of each graph shows mRNA levels of *mxc^mbn1^* larvae, mutant larvae heterozygous for *Toll^1-RXA^* mutation, and mutant larvae heterozygous for *imd^1^* mutation from left to right. Y-axis shows relative mRNA level of each target gene ((**a**) *Dro*, or (**b**) *CecA1*) to an endogenous control gene (*Rp49*). Significant differences between the groups were determined via one-way ANOVA for multiple comparisons (** *p* < 0.01, **** *p* < 0.0001, *n* = 3). The error bars indicate SEM. (**c**–**f**) DAPI-stained images of lymph glands (LGs) excised from male mature third-instar larvae. Shown are (**c**) normal control larvae, (**d**) *mxc^mbn1^* larvae, and (**e**,**f**) mutant larvae heterozygous for *Toll^1-RXA^* (**e**) and *imd^1^* (**f**) mutations, respectively. Scale bars: 100 µm. (**g**) Quantification graph indicates LG size of larvae with each genotype. Significant differences between the groups were determined using one-way ANOVA for multiple comparisons (*** *p* < 0.001, **** *p* < 0.0001). The red lines indicate mean LG size; the error bars indicate SEM.

**Figure 3 cells-14-00389-f003:**
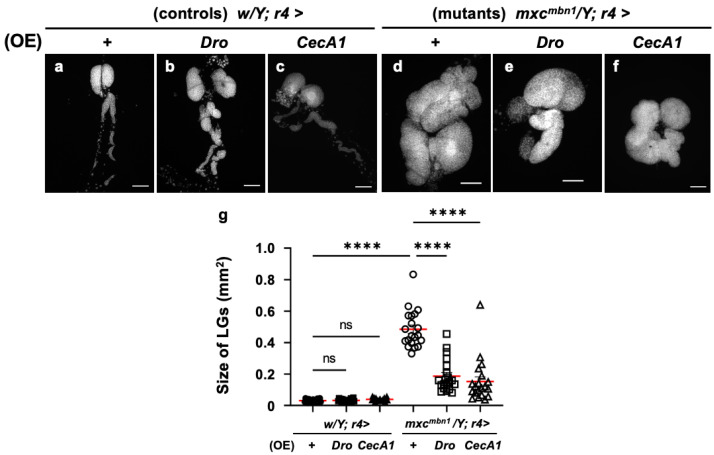
Observation of lymph glands (LGs) from *mxc^mbn1^* larvae and quantification of their size via induction of *Dro* or *CecA1* overexpression (OE) in a fat body (FB)-specific manner: (**a**–**f**) Fluorescence images of DAPI-stained LGs collected from mature third-instar larvae. (**a**) Pair of LGs from a normal control larva (*w/Y*; *r4>+*). (**b**) LG from control larvae with FB-specific overexpression of *Dro* (*w/Y*; *r4>Dro*) or (**c**) *CecA1* (*w/Y*; *r4>CecA1*). Pair of LGs from (**d**) *mxc^mbn1^* larva (*mxc^mbn1^/Y*; *r4>+*), (**e**) *mxc^mbn1^* larvae with FB-specific expression of *Dro* (*mxc^mbn1^/Y*; *r4>Dro*) or (**f**) *CecA1* (*mxc^mbn1^/Y*; *r4>CecA1*). Scale bars: 100 µm. (**g**) LG size quantification in larvae with FB-specific expression of *Dro* (*w/Y*; *r4>+* (*n* = 20), *w/Y*; *r4>Dro* (*n* = 20), *mxc^mbn1^/Y*; *r4>+* (*n* = 20), *mxc^mbn1^/Y*; *r4>Dro* (*n* = 20)), and *CecA1* ((*w/Y* (*n* = 20), *w/Y*; *r4>CecA1* (*n* = 20), *mxc^mbn1^/Y*; *r4>+* (*n* = 20), *mxc^mbn1^/Y*; *r4>CecA1* (*n* = 20)). Significant differences between the groups were determined using one-way ANOVA for multiple comparisons (**** *p* < 0.0001, ns: not significant). The red lines indicate the mean LG size; the error bars indicate SEM.

**Figure 4 cells-14-00389-f004:**
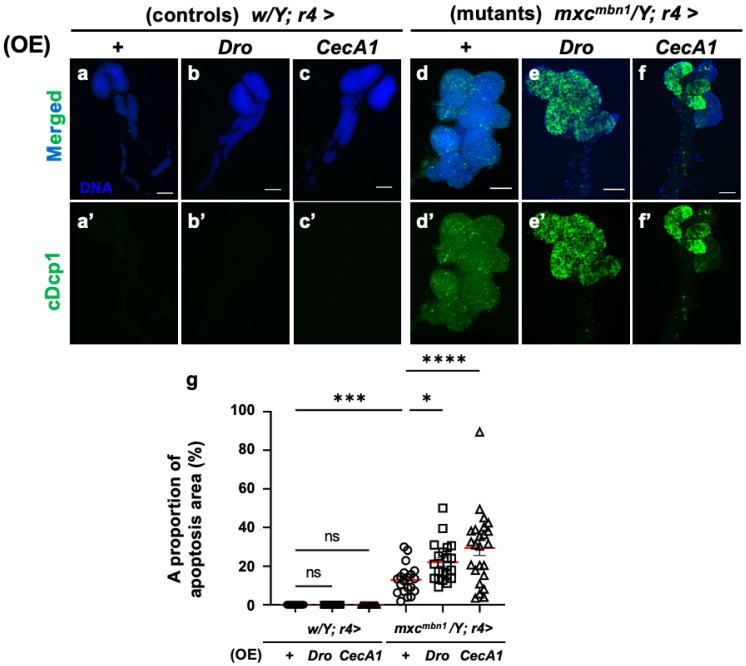
Observation and quantification of apoptosis areas in LGs of *mxc^mbn1^* larvae with FB (FB)-specific overexpression (OE) of *Dro* or *CecA1*: (**a**–**f**) Immunostaining of LGs with anti-cDcp1 antibody that recognizes apoptotic cells in LGs from the third instar-stage mature larvae. (**a**) Pair of LGs from control larvae (*w/Y*; *r4>+*). (**b**) Control larvae overexpressing *Dro* (*w/Y*; *r4>Dro*), or (**c**) *CecA1* (*w/Y*; *r4>CecA1*) specifically in FB. (**d**) Anterior lobes of pair of LGs from *mxc^mbn1^* larvae (*mxc^mbn1^/Y*; *r4>+*). (**e**) *mxc^mbn1^* larvae overexpressing *Dro* (*mxc^mbn1^/Y*; *r4>Dro*) or (**f**) *CecA1* (*mxc^mbn1^/Y*; *r4>CecA1*). Blue indicates DNA staining; green in (**a**–**f**) and (**a′**–**f′**) indicates anti-cDcp1 immunostaining signals. Scale bars: 100 µm. (**g**) Percentage of areas occupied by apoptotic cells in lobe regions of LGs from larvae with FB-specific *Dro* overexpression (*n* = 21 LGs from 11 larvae) or *CecA1* (*n* = 24 LGs from 12 larvae). Significant differences between the groups were determined using one-way ANOVA for multiple comparisons (* *p* < 0.05, *** *p* < 0.001, **** *p* < 0.0001, ns: not significant). Red line indicates the mean percentage of apoptosis. The error bars indicate SEM.

**Figure 5 cells-14-00389-f005:**
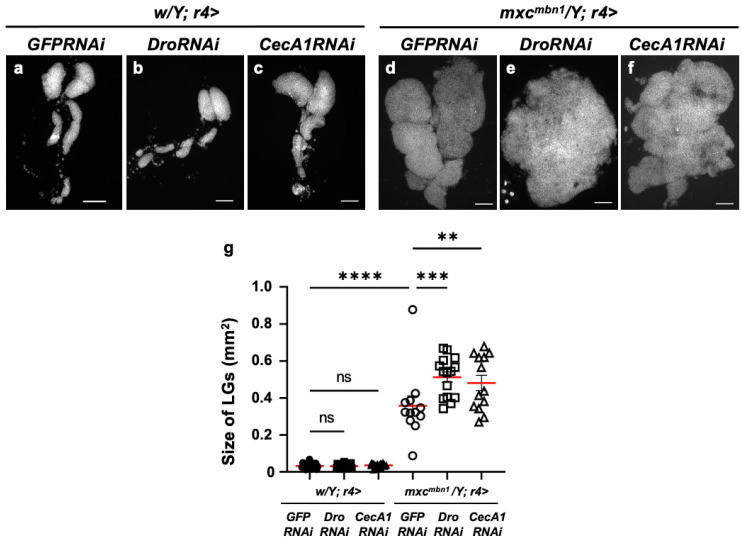
Quantification of LG sizes in *mxc^mbn1^* larvae with FB-specific knockdown of *Dro* or *CecA1:* (**a**–**f**) DAPI-stained images of LGs from mature third-instar larvae. (**a**–**c**) LGs expressing dsRNAs against mRNAs for (**a**) *GFP* (*w/Y*; *r4>GFPRNAi*) (control), (**b**) *Dro* (*w/Y*; *r4>DroRNAi*), or (**c**) *CecA1* (*w/Y*; *r4>CecA1RNAi*) specifically in FB are shown. (**d**–**f**) LGs expressing dsRNAs against (**d**) *GFP* in FB of *mxc^mbn1^* larvae (*mxc^mbn1^/Y*; *r4>GFPRNAi*), (**e**) *Dro* (*mxc^mbn1^/Y*; *r4>DroRNAi*) or (**f**) *CecA1* (*mxc^mbn1^/Y*; *r4>CecA1RNAi*). Scale bars: 100 µm. (**g**) Quantification graphs of the LG size in larvae of each genotype have. LG size of larvae with *DroRNAi* (*n* = 15 LGs from 8 larvae) and *CecA1RNAi* (*n* = 13 LGs from 7 larvae). Significant differences between the groups were determined using one-way ANOVA for multiple comparisons (** *p* < 0.01, *** *p* < 0.001, **** *p* < 0.0001, ns: not significant). The red lines indicate mean LG size. The error bars indicate SEM.

**Figure 6 cells-14-00389-f006:**
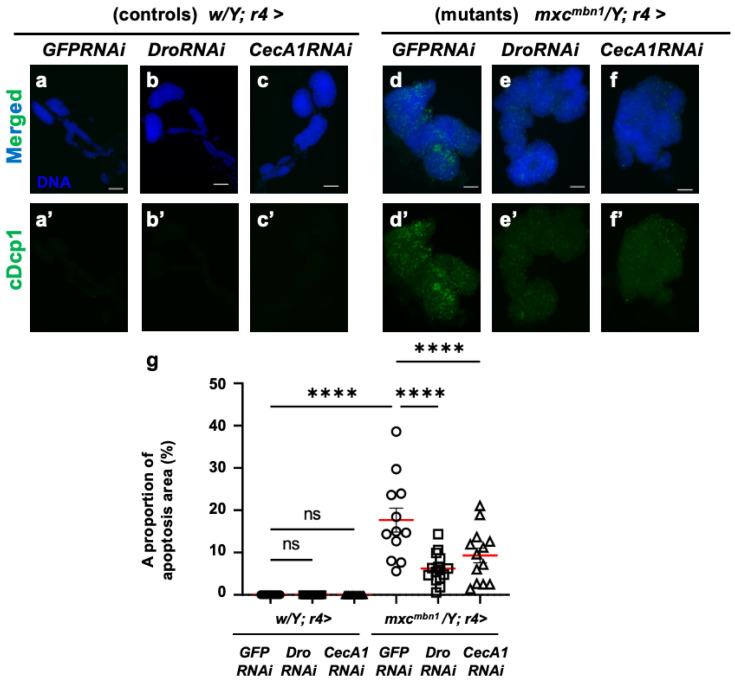
Apoptosis observation and quantification in LGs of *mxc^mbn1^* larvae with FB-specific knockdown of *Dro* or *CecA1*: (**a**–**f**) Immunostaining of LGs with anti-cDcp1 antibody that recognizes apoptotic cells. LGs expressing dsRNA against mRNAs for (**a**) *GFP* (*w/Y*; *r4>GFPRNAi*), (**b**) *Dro* (*w/Y*; *r4>DroRNAi*), or (**c**) *CecA1* (*w/Y*; *r4>CecA1RNAi*) specifically in FB are shown. (**d**–**f**) LG expressing dsRNA against (**d**) *GFP* specifically in FB of *mxc^mbn1^* larvae (*mxc^mbn1^/Y*; *r4>GFPRNAi*), (**e**) *Dro* (*mxc^mbn1^/Y*; *r4>DroRNAi*), or (**f**) *CecA1* (*mxc^mbn1^/Y*; *r4>CecA1RNAi*) are shown. Blue indicates DNA staining; green in (**a**–**f**) and (**a′**–**f′**) indicates anti-cDcp1 immunostaining signals. Scale bars: 100 µm. (**g**) Graphs indicate percentage of apoptotic cells in LG lobe regions of larvae with FB-specific depletion of *Dro* (*n* = 15 LGs from 8 larvae) or *CecA1* (*n* = 13 LGs from 7 larvae). Significant differences between the groups were determined using one-way ANOVA for multiple comparisons (**** *p* < 0.0001, ns: not significant). The red line indicates the mean percentage of apoptosis. The error bars indicate SEM.

**Figure 7 cells-14-00389-f007:**
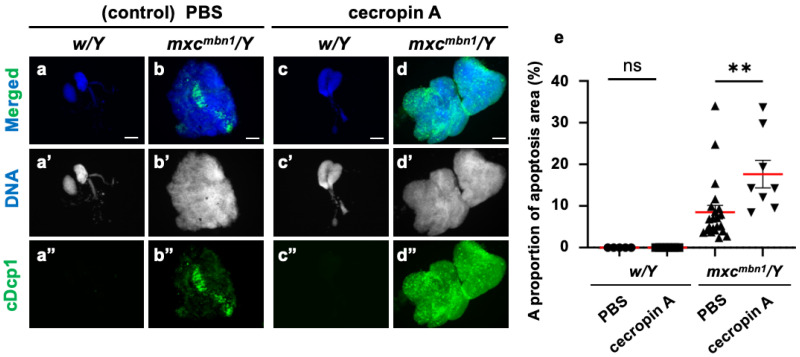
Apoptosis area quantification in *mxc^mbn1^* larvae LGs after synthetic cecropin A peptide injection: (**a**–**d**) Immunostaining of LGs in control (**a**,**c**) and *mxc^mbn1^* (**b**,**d**) larvae with anti-cDcp1 antibody that recognizes apoptotic cells. Third-instar larvae injected with PBS (control; (**a**,**b**)) or synthetic cecropin A (**c**,**d**) dissolved in PBS. Green in (**a”**–**d”**) indicates signal of anti-cDcp1 immunostaining, and blue (white in (**a′**–**d′**)) indicates DNA staining. Scale bars: 100 µm. (**e**) Quantification graphs indicate percentage of apoptotic cells in LG lobe regions after injecting PBS (*n* = 5 LGs from 3 *w/Y* and *n* = 22 LGs from 11 *mxc^mbn1^/Y* larvae), and cecropin A (*n* = 7 LGs from 4 *w/Y* and *n* = 8 LGs from 4 *mxc^mbn1^/Y* larvae). Significant differences were determined using one-way ANOVA for multiple comparisons (** *p* < 0.01, ns: not significant). The red line indicates mean percentage of apoptosis. The error bars indicate SEM.

**Figure 8 cells-14-00389-f008:**
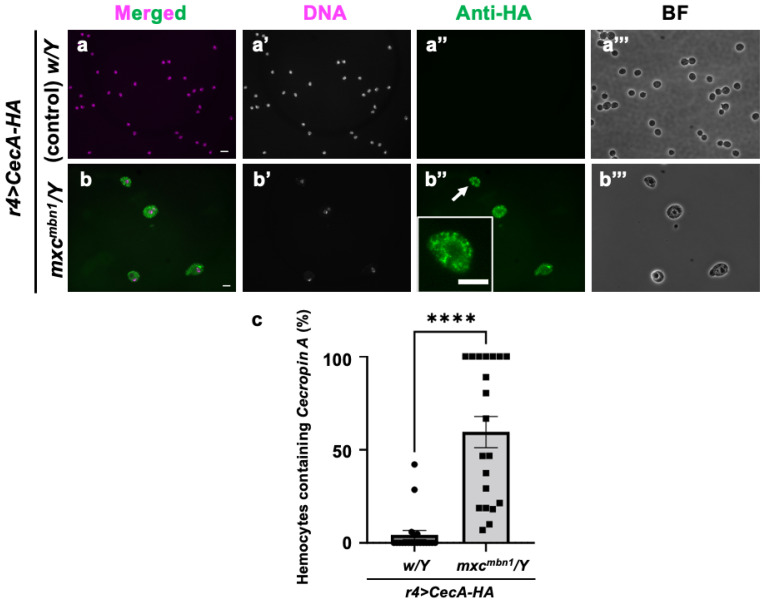
Observation of circulating hemocytes containing HA-tagged Cecropin A produced in the FB in control and *mxc^mbn1^* larvae: (**a**,**b**) Merged images of anti-HA immunostaining and DNA staining of circulating hemocytes in normal (*w/Y*; *r4>CecA1-HA*) (**a**) and *mxc^mbn1^* larvae (*mxc^mbn1^/Y*; *r4>CecA1-HA*) (**b**) expressing Cecropin A-HA in the FB. Green in (**a**,**b**,**a″**,**b″**), fluorescence of anti-HA immunostaining; magenta in (**a**,**b**), DNA staining (white in **a′**,**b′**). Magnified image of hemocyte indicated with an arrow is presented in insets in (**b″**). Bright-field (BF) images (**a‴**,**b‴**). Scale bars: 10 µm. (**c**) Percentages of hemocytes harboring HA-tagged Cecropin A in control and *mxc^mbn1^* larvae. Significant differences were determined using Welch′s *t*-test (**** *p* < 0.0001, *n* = 20). The error bars indicate SEM.

**Figure 9 cells-14-00389-f009:**
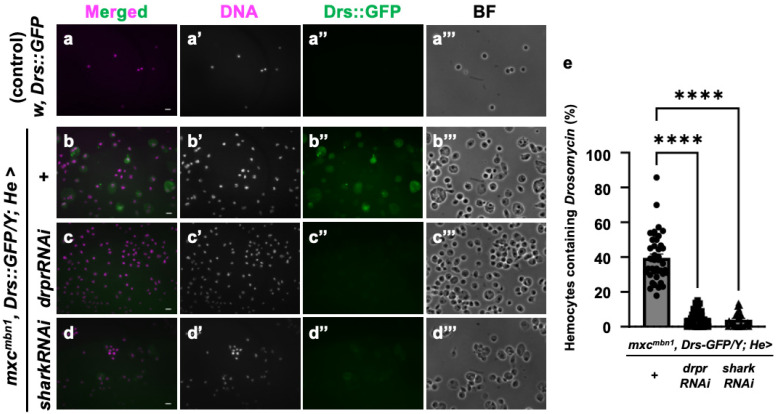
Observation and quantification of hemocytes in which GFP-tagged Drosomycin was incorporated in control and *mxc^mbn1^* larvae: (**a**,**b**) GFP fluorescence of circulating hemocytes to detect GFP-tagged Drosomycin (**a**,**b**), induced in the FB of control (*w*, *Drs::GFP/Y*) (**a**) and *mxc^mbn1^ (mxc^mbn1^*, *Drs::GFP/Y*) (**b**) larvae. (**c**,**d**) GFP fluorescence indicating GFP-tagged Drosomycin in circulating hemocytes of mutant larvae with hemocyte-specific knockdown of *draper* (*mxc^mbn1^*, *Drs::GFP/Y*; *He>drprRNAi*) (**c**), or *shark* (*mxc^mbn1^*, *Drs::GFP/Y*; *He>sharkRNAi*) (**d**). Circulating hemocytes with GFP-tagged Drosomycin (Drs::GFP) are colored in green in (**a**–**d**,**a″**–**d″**). DNA is magenta in (**a**–**d**) (white in (**a′**–**d′**)). Bright-field (BF) images (**a‴**–**d‴**). Scale bars: 10 µm. (**e**) Percentages of hemocytes with GFP-tagged Drosomycin in control and *mxc^mbn1^* larvae. X-axis from left to right: control larvae expressing GFP-tagged Drosomycin under its promoter (*w*, *Drs::GFP/Y* (*n* = 374 hemocytes (6 larvae)), *mxc^mbn1^*, *Drs::GFP/Y* (*n* = 1021 (8)), *mxc^mbn1^*, *Drs::GFP/Y*; *He>drprRNAi* (*n* = 2098 (8)), and *mxc^mbn1^*, *Drs::GFP/Y*; *He>sharkRNAi* (*n* = 1193 (6)). Significant differences were determined using one-way ANOVA for multiple comparisons (**** *p* < 0.0001). The error bars indicate SEM.

**Figure 10 cells-14-00389-f010:**
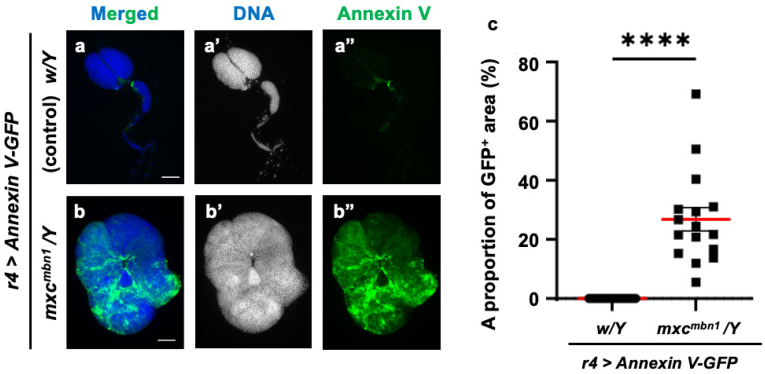
Detection of phosphatidylserine (PS) exposed on cell surface of lymph gland (LG) tumors in control and *mxc^mbn1^* larvae: (**a**,**b**) DAPI-stained fluorescence images of LGs from larvae at the third instar stage: (**a**) normal control; (**b**) *mxc^mbn1^* mutant. Blue in a,b (white in (**a′**,**b′**)) indicates DNA staining and green in (**a**,**b**) and (**a″**,**b″**) indicates Annexin V-GFP signal. Scale bars: 100 µm. (**c**) Quantification graph indicating percentage of GFP fluorescent regions in LGs, indicative of Annexin V binding. Significant differences were determined using Welch’s *t*-test (**** *p* < 0.0001, *n* = 16). The red line indicates mean percentage. The error bars indicate SEM.

**Figure 11 cells-14-00389-f011:**
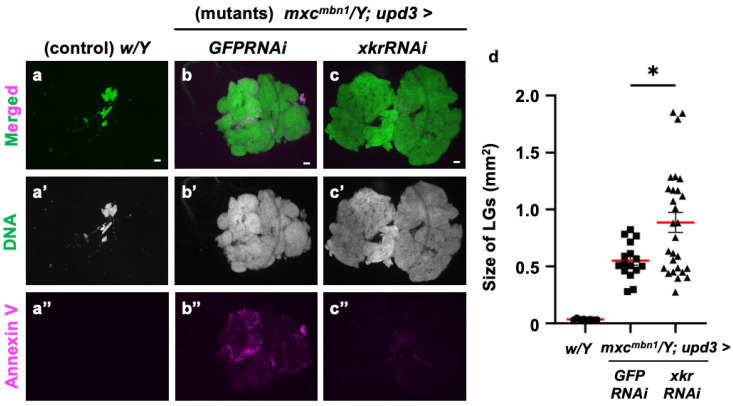
Loss of PS on the surface of LG cells via Xkr scramblase knockdown and its influence on LG hyperplasia in *mxc^mbn1^* larvae: (**a**–**c**) DAPI-stained anterior lobes and fluorescence indicating Alexa 594-Annexin V binding to PS on the LG lobes in normal control (*w/Y*) larvae (**a**), *mxc^mbn1^* larvae with the ectopic expression of control dsRNA in the medulla zone in primary lobes of the LG (*mxc^mbn1^*/*Y*; *upd3>GFPRNAi*) (**b**), *mxc^mbn1^* larvae with the depletion of *xkr* mRNA (*mxc^mbn1^/Y*; *upd3>xkrRNAi*), and (**c**) larvae at the third instar stage. DNA is stained in blue in (**a**–**c**) (white in (**a′**–**c′**)), and Alexa594-Annexin-V is in magenta in (**a**–**c**,**a″**–**c″**). Scale bars: 100 μm. (**d**) Quantification of the LG size of *mxc^mbn1^* larvae with *xkr* depletion in LG tumor cells. The average LG size was calculated among the controls (*w/Y*) (*n* = 9 LGs (5 larvae)), *mxc^mbn1^/Y*; *upd3>GFPRNAi* (*n* = 16 (8)), and *mxc^mbn1^/Y*; *upd3>xkrRNAi* (*n* = 28 (14)). Significant differences were determined using one-way ANOVA for multiple comparisons (* *p* < 0.05). The red lines indicate the mean percentage of apoptosis or the mean LG size. The error bars indicate SEM.

## Data Availability

The datasets generated and/or analyzed in this study are available from the corresponding author upon reasonable request.

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
