# Peer review of "Anti-Tumor Effects of Cecropin A and Drosocin Incorporated into Macrophage-like Cells Against Hematopoietic Tumors in Drosophila mxc Mutants"

_cells, 2025, doi:10.3390/cells14060389_

Round 1

Reviewer 1 Report

Comments and Suggestions for Authors

The authors utilized a Drosophila lymphoma model induced by mxc gene mutation to investigate the antitumor effects of antimicrobial peptides (Cecropin A and Drosocin). The study demonstrated that Cecropin A and Drosocin are expressed in the fat body and secreted into the hemolymph, with their anti-lymphoma activity being associated with phosphatidylserine (PS) on the tumor cell surface. These peptides do not affect the proliferation of healthy cells but induce apoptosis in tumor cells.

However, the language of the manuscript does not conform to academic writing standards, particularly in the results and abstract sections. Excessive details on experimental procedures and data obscure the key findings and overall logical flow of the study. It is recommended to remove direct descriptions of result figures in the main text and instead summarize the experimental results concisely, emphasizing their biological significance. Additionally, the green fluorescence images in the result figures should be presented in color, and references to P-values in the main text should be omitted, with the statistical significance indicated in the figure legends instead.

Comments on the Quality of English Language

It is recommended to seek assistance from a native English-speaking cell biologist to refine the language of the manuscript.

Author Response

Reviewer 1

Comments and Suggestions for Authors

However, the language of the manuscript does not conform to academic writing standards, particularly in the results and abstract sections. Excessive details on experimental procedures and data obscure the key findings and overall logical flow of the study. It is recommended to remove direct descriptions of result figures in the main text and instead summarize the experimental results concisely, emphasizing their biological significance.

(Response) First, the authors appreciate this reviewer for his/her careful reading and for providing thoughtful comments.

Upon the request of reviewer 1, to clarify the logical flow of the entire study and make the main findings more straightforward, we revised more than 150 sentences to make the text, mainly the Results and Abstract sections, concise so that our key findings are highlighted. Then, we asked the MDPI English proofreading services (English ID: english-90562) to check the quality of the English by experienced, native English speaking editors and asked for revisions to improve the quality of the English. We correct all errors in grammar and edit the manuscript to a level suitable for a scientific journal as instructed. Please see the revised manuscript with tract changes. Please see the certificate uploaded with the manuscript.

Additionally, the green fluorescence images in the result figures should be presented in color, and references to P-values in the main text should be omitted, with the statistical significance indicated in the figure legends instead.

(Response) As requested, we changed the color used to display green fluorescence from white to green in Fig. 1a’-d’, Fig. 3a’-f, Fig. 5a’-f, Fig. 6a”-d”, Fig. 7b”, Fig. 8a”-d”, and Fig. 9a”, b”. In addition, we also changed the color of the red fluorescence images in Fig. 10a”-c” from white to magenta.

We also removed all references of p-values from the main text and instead added the statistical significance in the figure legend of each figure (figure legends in Figs. 1-11).

Reviewer 2 Report

Comments and Suggestions for Authors

The authors describe tumour-suppression effects by two anti-microbial peptides (AMPs), Drosocin and Cecropin A in a Drosophila larval lymph gland tumour model. Key findings of this study are that these two AMPs are expressed by the fat body downstream of the innate immune pathways. The authors demonstrate that the two AMPs exhibit growth reducing activity of lymph gland tumours by inducing apoptosis rather than blocking proliferation. They show that Cecropin A is taken up by macrophages and that Cecropin A from other insect elicits tumour suppressive activity when injected into the larval tumour model. Towards a mechanism for tumour cell detection, they demonstrate that phosphatidylserine is present on lymph gland tumour cells and propose that AMPs may act on cells with high PS levels on the outer membrane leaflet and induce apoptosis. Finally the authors show that the uptake of the AMP Drosomycin requires factors involved in phagocytosis, but the mechanism of the uptake remains an open question. This is a high quality experimental work presented in a well written manuscript. The paper fits well into the special issue as it suggests that the two AMPs, in particular Cecropin A may be a lead for anti-tumour compounds. 

I suggest a few issues that the authors may want to take into account before publishing. 

Overall the manuscript is very well written and I did not detect any typos. However, at times, it occurred to me that some more background information would make this paper more accessible to the interested non-expert (fly) audience. Also it would be interesting to comment on whether the tumour suppressing activity of the two novel AMPs is/would be expected to also act on other tumour models in the fly?

(1) The data presented as Supplementary Figure S1 should be moved into the main results section as they are important to setting the stage for the tumour model.

(2) The figure annotations/labels in Figs. 2 a-f and 3a-f could be improved to clearly indicate what is mutant and what is over expression.

(3) The rationale of using anti-cDCP1 staining as a marker for apoptotic cells is not explained here. More common would be TUNEL staining. Please provide additional evidence (or reference) that anti-cDCP1 is a reliable apoptosis marker. 

(4) The authors argue that proliferation is not affected by the AMPs by anti-phosHistone 3 staining. Again, the assay may be established in this field

some technical issues:

(1) the quantification of the extent of hyperplasia or tumour size should be explained in more detail. It is not a trivial task to determine tumour size as two-dimensional entity (in mm2). This reviewer would urge the authors to provide a more convincing description of the method or use a distinct method, which would take the 3D shape of LG tumours into account. 

Author Response

Reviewer 2

Comments and Suggestions for Authors

I suggest a few issues that the authors may want to take into account before publishing.

Overall the manuscript is very well written and I did not detect any typos. However, at times, it occurred to me that some more background information would make this paper more accessible to the interested non-expert (fly) audience. Also it would be interesting to comment on whether the tumour suppressing activity of the two novel AMPs is/would be expected to also act on other tumour models in the fly?

(Response) First, the authors appreciate this reviewer for his/her careful reading and for providing thoughtful comments.

We agree that it is very interesting to investigate whether the anti-tumor properties of these two novel AMPs can be exerted on other types of tumor models in Drosophila. Several experiments regarding this issue are currently underway. We hope that we will submit the data to our future papers.

(1) The data presented as Supplementary Figure S1 should be moved into the main results section as they are important to setting the stage for the tumour model.

(Response) According to the reviewer’s request, we relocated the previous Fig. S1 to the main results, designating it as the new Fig. 2. The numbers of the subsequent figures were each increased by one.

(2) The figure annotations/labels in Figs. 2 a-f and 3a-f could be improved to clearly indicate what is mutant and what is over expression.

(Response) As requested, we have added annotations to clearly indicate which images represent LGs in control or the mxc mutant larvae with or without the overexpression of the AMP in Figs. 3 (previous 2) a-f and 4 (previous 3) a-f. 

(3) The rationale of using anti-cDCP1 staining as a marker for apoptotic cells is not explained here. More common would be TUNEL staining. Please provide additional evidence (or reference) that anti-cDCP1 is a reliable apoptosis marker.

(Response) At the reviewer’s request, we provided additional descriptions and references in Materials and Methods 2.5 (lines 177-179), indicating that anti-cDCP1 is a reliable apoptosis marker in the lymph gland.

(4) The authors argue that proliferation is not affected by the AMPs by anti-phosHistone 3 staining. Again, the assay may be established in this field.

(Response) In response to the reviewer’s request, we provide additional descriptions and references indicating that the immunostaining of the tissue using anti-phospho-Histone 3 revealed mitotic cells within the tissues, as detailed in Materials and Methods 2.5 (lines 179-181).

some technical issues:

(1) the quantification of the extent of hyperplasia or tumour size should be explained in more detail. It is not a trivial task to determine tumour size as two-dimensional entity (in mm2). This reviewer would urge the authors to provide a more convincing description of the method or use a distinct method, which would take the 3D shape of LG tumours into account. 

(Response) According to the reviewer’s request, we added a more detailed explanation in Materials and Methods 2.4 (lines 170-172) as follows: “The fixed LG specimen was mounted in a mounting medium (Vector Laboratories, CA, USA) under a cover glass, and mildly flattened under constant pressure using an apparatus so that the tissue spread out into thin cell layers with a constant thickness [26].” (lines 170-172). Then, we measured the size of each DAPI-stained LG on the acquired fluorescence image using Image J software. We had previously established this procedure to quantify the tumor size [26], and after that, we used the same method in our previous studies [26, 32, 35, 37, 39, 51]. We cited one of our previous publications in the text.

Round 2

Reviewer 1 Report

Comments and Suggestions for Authors

It is fine for publication.